# Reflections on Digital Nomadism in Spain during the COVID-19 Pandemic—Effect of Policy and Place

Juan Parreño-Castellano *, Josefina Domínguez-Mujica and Claudio Moreno-Medina

Department of Geography, University of Las Palmas de Gran Canaria, 35003 Las Palmas de Gran Canaria, Spain
* Correspondence: juan.parreno@ulpgc.es

**Abstract:** To the extent that digital capitalism and globalization processes have been developing, the arrival of digital nomads has grown in Spain. With the pandemic, this mobility was affected to a lesser magnitude than other types of flow. In this context, this paper deals with the study of the characteristics of digital nomads and the policies developed to attract them during the health crisis. With these objectives, the research, in relation to digital nomads, has been carried out based on the analysis of different virtual platforms, social networks, portals of collaborative workspaces and specialized events. At the same time, with respect to policies, the study has been focused on visa policy and on the actions developed by destinations to boost this type of mobility. The results obtained indicate, on the one hand, that it is not a flow of privileged people but a mobility like that of tourism related to the difference in international income. On the other hand, these results point out that the consolidation of digital nomadism during the pandemic is associated to tourism policies carried out by destinations, actions that have not valued the lack of sustainability of digital nomadism.

**Keywords:** digital nomadism; workation migration; remote working; Spain; policy; pandemic

## 1. Introduction

The mobility paradigm, which has been so successful in the field of social sciences since the turn of the century, has been reinforced by digital nomadism, a symbol of flux, hybridity, and mobility in a globalizing world [1]. Digital capitalism has contributed to create new forms of organization, in which work and place have lost weight as identity elements, and delocalization, deregulation and leisure have gained importance. In this socio-productive context, new lifestyles linked to residential relocation are appearing, such as digital nomads (those digital workers who make mobility the structuring axis of their way of life) and remote workers or workation migrants (those who have voluntarily relocated their residence temporarily in favor of some place, where they carry out a lifestyle model compatible with leisure, without this implying hyper-mobile behavior). Both carry out their work activities taking advantage of the new telecommunication technologies and combining this situation with a desire to explore the world or to enjoy a quality of life linked to outdoor recreation. Through strategic uses of agency within new structural constraints, they transform postmodern conditions of uncertainty and constant change into new opportunities of work and leisure within a global society [2].

The practice of digital nomadism began to gain importance in the first decades of the 21st century, while remote work dates back to the 1970s. Both modalities have shown great strength in the last decade and became a growing trend with the COVID-19 pandemic. Paradoxically, the pandemic, which contained many of the international flows of travelers, favored residential relocation and nomadism linked to teleworking. In this context, it is important to analyze the change given to place. As opposed to the meaning it had in the Fordist productive organization when it represented a defining factor of identity, a space for living and working together, in post-Fordist capitalism, we must speak of a set of places linked to production, leisure or mobility, as well as the digital space itself in which nomads and remote workers develop [3].

Thus, digital nomads (DNs onwards) (a term that includes remote workers) maintain very weak emotional ties with the areas in which they live since their personal identity is determined by their own mobility and not by their roots. They prefer certain places with fast on-line connectivity, a pleasant climate, low cost of living and an enjoyable natural and cultural environment to share experiences, manage relationships and find elements of "sameness" and stability. Consequently, an international market of destinations for DNs has been developed. It has involved a growing number of cities and tourist spaces that compete thanks to the public and private initiatives trying to position them on that international market. However, most research on digital nomadism has paid little attention to the idiosyncrasies of these destinations or to the policies developed to promote them [4]. To fill this gap, we propose a geographical approach to the places and initiatives for the attraction of these people to Spain and, specifically, to the Canary Islands.

The presence of DNs in Spain is recent. Since the middle of the previous decade, a considerable number of them began to arrive in its largest cities and many of its tourist areas. However, this evolution was slowed down in 2020 as a result of the pandemic. The lockdown (March to May 2020) caused a sharp drop in figures of arrival, given the limitations on mobility and the closure of accommodation establishments. Once the period of confinement was over, with the gradual de-escalation on restrictions to entry into Spain, the arrival of DNs resumed. For example, according to Homelike (2022), Europe's leading long-stay temporary accommodation rental platform, demand for housing by DNs in Spain grew by 291% in August 2021, compared to the same period in 2020. Similarly, in the Canary Islands, according to the Sociedad de Promoción Económica de Gran Canaria (Society for Economic Promotion of Gran Canaria) (SPEGC), it is estimated that 87,000 digital nomads will visit the archipelago in 2022, double that of the previous year.

This growing importance of digital nomadism in Spain makes it necessary to characterize the geography of destinations. Consequently, the first objective of this study is to recognize the level of implementation of digital nomadism in the main Spanish reception places, that is, the geography of the reception places, from the information provided by the web portals created by and for the DN communities, a valuable source used by many other researchers [5–7]. This makes it possible to identify the dimension of digital nomadism from the perspective of the assessment of the destinations according to the opinions of the DNs themselves. We also use data on the supply of accommodation and workspaces where they carry out their activities, which indicates the level of specialization of the destinations in relation to the facilities that attract these workers (apartments, vacation homes, co-working spaces, co-living spaces, etc.). In short, knowing the destinations allows us to understand more precisely the mobility of DNs based on the characteristics of the reception spaces (see Figure 1).

The second objective of the research is to find out what the DNs who have arrived in Spain, and particularly in the Canary Islands, are like, and to characterize the relationship between the profile of the DNs and the geography of the host destinations. Much of the literature states that most of these independent workers always face the risk of sliding into "precarious work" situations since they do not have access to the wider safety net often provided by a business to its full-time employees [8,9]. For this reason, we wanted to delve deeper into the socioeconomic status of the DNs.

Finally, as some authors have pointed out, during the pandemic, DNs continued to defend the alternative of staying nomads, as opposed to the theories of "going back 'home', searching for (??) security" [10], sharing in social network strategies to face restrictions to mobility and the best practices on how to plan a trip [11]. This fact demonstrated their greater capacity for resilience in the face of (im)mobility and, for this reason, after the confinement, the reception spaces for DNs launched some public and private initiatives to boost their arrival. Therefore, the third objective of this paper is to analyze the national, regional and local policies carried out, with special attention to the Canary Islands. The purpose is to understand the tourist character of the mobility of DNs in Spain better by analyzing the public actions carried out and their role in the consolidation of this mobility.

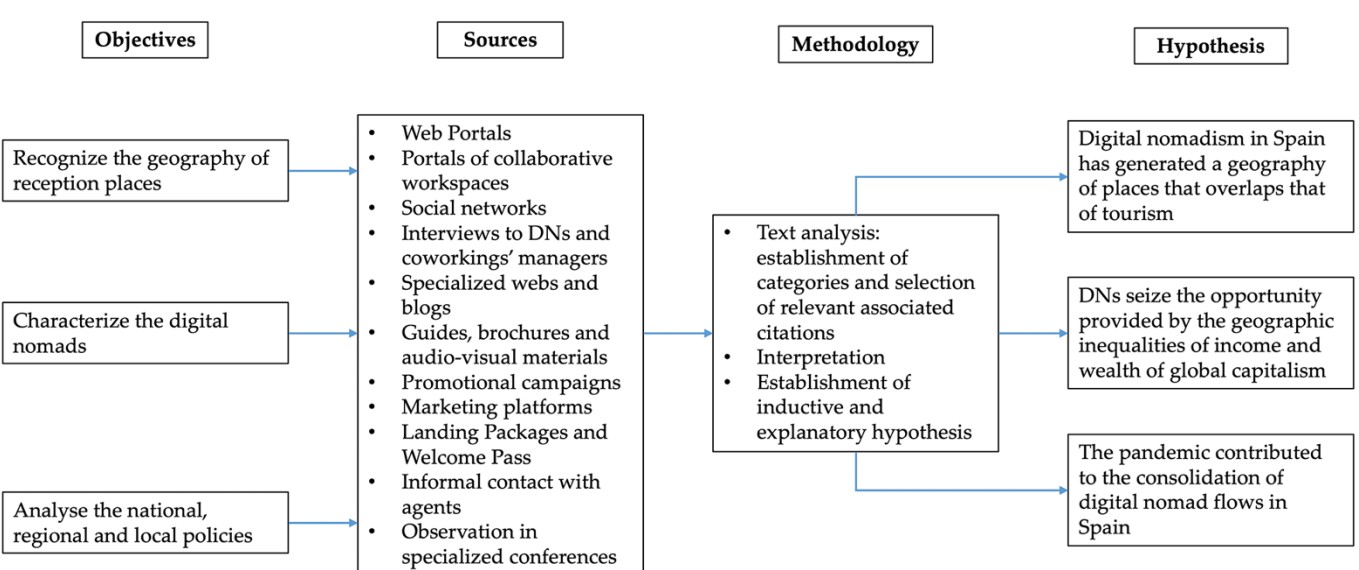

**Figure 1.** Outline of the research process. Source: The authors.

To address the above-mentioned objectives, we have analyzed and interpreted the information gathered by formulating three inductive hypotheses, which will need to be confirmed with other case studies. This is the first step towards the elaboration of general theories. These are explanatory hypotheses that emerge from the qualitative analysis carried out. Firstly, digital nomadism in Spain has generated a geography of places, which is not very different from that created by tourism development, because the pull factors are similar. In this regard, it must be remembered that Spain and, specifically, the Canary Islands, for more than one-half of a century, have been main destinations for international tourism in Europe. In 2019, for example, Spain welcomed 83.5 million foreign tourists, and The Canary Islands welcomed 13.1 million, according to INE (National Statistics Institute). Therefore, the preexisting geography of Spanish tourist destinations is, to a large extent, that which welcomes DNs. Therefore, the use of the tourist accommodation offer by the DNs suggests that it does not differ from that of the tourists themselves.

Secondly, without ignoring the internal diversity in the flows of DNs, the analysis of the data has led us to interpret that we are facing a mobility based on the existence of disparities in living standards between countries and between regions. Contrary to the view that DNs are people with high purchasing power, most of those who arrive do so by taking advantage of this income differential between their countries of origin (United Kingdom, Denmark, Sweden, Iceland, Switzerland, the Netherlands, Germany, etc.) and the places of reception in Spain. In other words, they take advantage of the opportunity offered by the geographical inequalities of income and wealth of global capitalism.

The last hypothesis resulting from the analytical induction is that the pandemic contributed to the consolidation of DN flows in Spain by replacing, in part and temporarily, international tourists. With the onset of the pandemic, tourist arrival figures dropped considerably, with only 18.9 and 3.7 million in 2020, in Spain and the Canary Islands, respectively. However, the arrival of DNs showed greater resilience, which is why public actions to attract these flows multiplied and have ended up consolidating this mobility.

With these objectives, the article is structured as follows: (i) a section reviewing the mobility of DNs in times of pandemic, from a theoretical perspective; (ii) a part specifying the methodology used; (iii) a piece providing results, on the one hand, for the characteristics of DNs who arrived in Spain and, specifically, in the Canary Islands and, on the other hand, on the geography of the places that have hosted them; (iv) an epigraph oriented to the analysis of the main policy actions carried out to promote the arrival of DNs in the Canary Islands; and (v) the concluding remarks. This section synthesizes the interpretations that led

us to the hypotheses and presents some reflections on the mobility–work–tourism–policy nexus and the sustainability of destinations.

## 2. Background: A Literature Review

### 2.1. Digital Nomadism in the Context of Global Capitalism

The 1970s marked the progressive breakdown of the Fordist system. For many, the breaking point from which a new stage began to emerge was in 1973, which has been called post-Fordist, post-industrial or the phase of the information society or global capitalism. These concepts define, from an economic point of view, a profound structural change due to the globalization of production and markets, and the rapid and intense renewal of companies' competitive strategies under flexible modes of organization and volatile financial capital [12,13]. From a technological point of view, this economic restructuring coincides with a new revolution based on digital information and communication; and, from a political point of view, in a context of multipolarity, with a growing liberalization of the regulatory functions that States had been performing [14]. To these characteristics must be added, from a sociological perspective, the emergence of transnational actors and new migration waves in a context of reflexive forms of social organization and identity.

These new forms of identity and sociality interrelate with global processes of hyper-mobility, digitalization, subjectivity and reflexivity [15], of which digital nomadism is a clear exponent that can be understood as a process of self-construction through which the reflexive project of the self turns into an explicit form of labor under post-Fordist flexible capitalism [16]. Rather than a challenge to the system [17], DNs provide just one example of the increasingly dominant narratives of work in which flexibility and hyper-productivity have become normative [18]. Therefore, digital nomadism is explained by the role of human work in the logic of the neoliberal order, as labor becomes increasingly 'liquid' [19], forcing a new definition of work [20] in which the traditional dichotomy between production and consumption is blurred and the values of fluidity and entrepreneurialism emerge. This has meant that young people, who have joined the labor market and productive activity in recent years have developed ductile work behavior, although without being free from precariousness. This is confirmed by the uncertain labor future of DNs or their frequent job changes, which they combine with periods of self-employment, more as an economic coping strategy than as an empowering life dream [21].

Some of the characteristics of globalization that contribute to the development of digital nomadism are summarized by Alonso-Calero and Cano-García [22]: the development of portable computational and telecommunication means and global connectivity through the internet; the growth of the digital service economy that allows the realization of skilled jobs that are sold through the network; the implementation of work organization models by objectives or tasks that do not require presence in the physical office and favor delocalized work and flexible schedules; the lowering of barriers to the mobility of people, temporary residence and the development of means of transportation and interconnectivity; the growth of collaborative economy models that facilitate the dissemination of information and the contracting of accommodation and work services; and the development of a business model aimed at these people, sponsored by public authorities, which encourages the creation of business networks. In brief, paraphrasing Bonneau and Aroles [23] (p. 21), it can be stated that "[...] digital nomadism goes hand in hand with the profound changes taking place in contemporary capitalism".

### 2.2. Broadening the Perspective of Lifestyle Mobilities: Digital Nomads and Workation Migrants

The term of DNs was first used by Makimoto and Manners in 1997 [24], and the literature reflections on these mobile people have multiplied since that date [25]. They have been portrayed by Reichenberger [26] as young professionals working solely in an online environment while leading a location-independent and often travel-reliant lifestyle. Turned into a lifestyle, mobility allows DNs to create their own life circumstances, defining their lives as an individualistic project of self-realization [17] facilitated by global access

to information and its infrastructure, more flexible work arrangements, a preference for travel, as well as adventure and work flexibility [27]. For this reason, other terms have been used, trying to give meaning to this new type of tourist-worker: knowledge nomads or Knowmad, emphasizing the creative and innovative nature of the activities carried out in the 3.0 society [28]; Neonomads, due to the reiterative migratory nature of some people [29]; Freeroaming [30]; Digital Gypsy, if mobility affects entire families; or Urban Bums, when they are conceptualized as a new precarious class [31].

From a sociodemographic perspective, most of them are male, white, between 26 and 36 years of age (millennials), coming from the Global North, from upper social strata or the middle class, and with a copiously visa-stamped passport [18,32]. They are highly educated workers with occupations such as that of programmers, bloggers, digital marketers, graphic designers, software engineers, financial traders, journalists, architects, etc. who maintain remote relationships with employers, clients, temporary collaborators or employees, according to their condition as freelancers, businessowners or employees. They normally travel alone or with a partner and, according to the period of stay in each place they visit, they can be classified as digital adventurers (with high mobility due to the very short-time stay), temporary digital nomads (short-term stay) or lifestyle remote workers and freelancers (long-term stay) whom we can also call workation migrants [33]. These DNs work maintaining a focus on connectivity and productivity, even in leisure, and merge their professional lives with personal and recreational time [5].

Nevertheless, many of their characteristics are common to those of the individuals who simply move driven by personal desires for a change in lifestyle, freedom of choice and self-fulfillment. To this respect, academic literature interpreting and reinterpreting the interrelations between different forms of human mobility has paid increasing attention to the patterns, effects and motivations of lifestyle migrations, whose limits and confluences with work and tourism have been well-theorized [34,35].

Therefore, the blurred boundaries between those who move for labor and lifestyle reasons are not unique to DNs. To the flows of relatively privileged individuals whose mobility practices are largely understood to be lifestyle-motivated, consumption-led and tourism-induced [36] must be added those whose reasons bridge between labor and lifestyle migrations. All these mobile individuals have contributed to transform mobility into the "order of the day" [37].

### 2.3. Tourism and Digital Nomadism: Moving the Focus from Agents to Places

Studies on digital nomadism from different disciplines (anthropology, sociology, economics, etc.) have been developing the concept of digital nomadism, approaching it from the perspective of the factors that contribute to the development of this form of mobility, whether by analyzing the advances in information technologies [6] or by interpreting its social, psychological, identity and subjectivity connotations [38]. To this, the reflections around places where the DNs develop their life and work are included. Special attention has been paid to the co-living and co-working spaces where they usually live, work and socialize, building sustainable relationships with people within a set digital nomad community [32].

Despite the fact that the lifestyle of DNs draws them to tourism destinations [39], very little has been written about the destinations beyond generalities: major cities, remote locations, tropical areas, natural environments, places to pursue hobbies or scenic places that are also affordable and easily accessible. As Bozzi [40] mentioned, DNs establish a geography intertwined between specific tourist flows and locations where normally tourists, retirement migrants, second homeowners, DNs and workation migrants cohabit. Consequently, some authors have pointed out the need to bring DNs to the tourism agenda to define these concepts in the scope of hospitality research [41] and to evaluate their potential for the tourism industry.

Many of these workers choose certain tourist destinations specialized in satisfying hobbies like surfing, hiking or skiing. Their main concerns are related to the availability of

short-term leasing to which purpose Airbnb lists are frequently of use to them. They also search for spaces where they can improve their work productivity (public libraries, cafés, shared offices and co-working spaces) and where they can find a "community" or a "digital nomads" hub [42]. Other important factors are the selection of places with fast internet connection, a medium-low cost of living and the presence of a community of like-minded people [17]. Therefore, researchers have begun to recognize the value of place in the DNs' lifestyle choice, but no geographical analysis has yet been carried out on these destinations nor on the urban–tourist and social impact that DNs produce in them. Although some studies have been carried out in Ibiza (Spain) [43]; Bali (Indonesia) [44]; Estonia [45]; Ljubljana (Slovenia), Leipzig and Berlin (Germany) and Prague (Czech Republic) [30]; Chiang Mai and Thailand as a whole [18,32,46] and the USA, UK, UAE, Australia and Singapore [47]—the only approach that relates the arrival of DNs to the processes of touristification in these destinations is that of McElroy [48], who alludes to gentrification processes that have taken place in Cluj (Romania) as a result of this new mobility phenomenon. In conclusion, studies on DNs have not yet moved the focus from agents to places.

## 3. Sources and Methodology

The academic literature on digital nomadism has focused on studying its sociological and labor implications. However, little has been written about the relationship between this mobility and the tourism destinations and infrastructures, on the one hand, and the institutional or private promotion policies, on the other. Consequently, sources related to places and to tourism destinations have not been sufficiently explored. In this study on digital nomadism, we have used different primary and secondary sources in order to address this approach, which is embodied in the three objectives mentioned above.

Firstly, knowledge of the Spanish geography of the destinations and their characterization has been obtained based on three types of sources: web platforms aimed at the group of DNs, portals of collaborative workspaces and social networks. In this way, the virtual media themselves, upon which digital nomadism is based, have become a source of research. Among the web platforms we highlight the information obtained from Nomad List (NomadList.com), one of the sources most used by DNs worldwide. This virtual space includes information on destinations that have hosted DNs since 2014 and its data come mainly from that provided by the interested parties themselves. It is an open-use platform but requires membership to access professional information. It classifies destinations according to multiple criteria, making it easy for its members to contact online work platforms and virtual groups of nomads through Discord. Some platforms, such as Airbnb or Homelike, with products specialized in long-term rentals were also used as a source of information, but the results were imprecise given the incipient nature of these products.

Regarding web portals for collaborative spaces, we have consulted Aecoworking, ACEC and CoworkingSpain. The first, Aecoworking (https://mobile.twitter.com/aecoworking, accessed on 5 December 2022) is an association of coworking spaces that operates throughout Spain and offers support and dissemination services to its members. ACEC (acecanarias.org, accessed on 5 December 2022) is a Canarian association of collaborative spaces, which aims to support its members, mediate with administrations and promote the recruitment of DNs. Finally, CoworkingSpain (coworkingspain.es, accessed on 5 December 2022) is an online platform of coworking spaces created in 2010 to offer a directory of coworking spaces that promote the culture of collaborative work. Through direct contact with these associations and platforms, we have been able to identify collaborative workspaces as well as obtain valuable information on the level of use of their facilities.

We also consulted and analyzed the information of some social networks created since 2015 to identify the characteristics of the destinations that the DNs themselves value. Table 1 includes a list of the networks consulted on Facebook (public and private groups, and local services) referring to December 2021.

**Table 1.** Social networks of digital nomads consulted.

| Name of the Group | Followers |
|---|---|
| Tenerife Remote Workers and Digital Nomads | 17,217 |
| Canary Islands Digital Nomads & Remote Workers | 11,900 |
| Digital Nomads Community Tenerife | 1500 |
| Tenerife Digital Nomads Community | 1100 |
| Tenerife Digital Nomad Community South and North | 5800 |
| Gran Canaria Digital Nomad | 17,100 |
| Expat Families Las Palmas de Gran Canaria | 848 |
| Accommodation for Digital Nomads in Gran Canaria | 2600 |
| Floworking La Oliva | 490 |
| Fuerteventura Digital Nomads | 284 |
| Lanzarote Digital Nomads & Remote Workers | 1400 |
| Enfoque Nómada | 574 |
| Nomad club | 884 |
| Dinero en sandalias | 9300 |

Source: The authors.

Secondly, the approximation of who the DNs are who visit these destinations was made from the mentioned social network publications (2021–2022). It should be noted that the intensity of DNs' mobility [49] and the lack of official records make it difficult to track them by means of traditional statistics. Therefore, a compilation of social media posts was used to better understand the characteristics of the group analyzed such as nationality, gender or profession; the travel model (duration, company), their needs and hobbies and their economic status from the type and price of accommodation required. Likewise, the commercial offers of companies and individuals from these social networks allowed us to delve deeper into their consumption patterns. The virtual information was completed with that provided through direct contact with some DNs and coworking space managers.

Finally, in relation to the third objective, the study of institutional actions to attract DNs, firstly, the different instruments of promotion and information that have been implemented in the case study were examined. For the public sphere, specialized websites, such as Nomad City Gran Canaria or Tenerife Work and Play, guides, brochures and specific audiovisual material produced by the promotional brands of the Councils of Gran Canaria and Tenerife were consulted. Also, the tracking of promotional campaigns of the Government of the Canary Islands, Cabildo de Fuerteventura (Fuerteventura Council), Cabildo de Tenerife (Tenerife Council) and Ayuntamiento de La Oliva (La Oliva Town Hall) and public collaborative actions with marketing platforms such as Live and Work Anywhere was carried out.

Secondly, the Landing Packages and Welcome Pass that some island institutions have developed in order to make it easier for DNs to settle on the islands were analyzed. Finally, we approached the entrepreneurial ecosystem through contact with agents and carried out an observation and informal contact with the participants in the People Conference (the largest congress for digital nomads in Europe). This allowed us to appreciate the business relationships generated by the arrival of DNs.

Given that most of the information corresponds to documents published virtually, posts on social networks and transcribed oral testimonies, we resorted to the analysis of texts, for which we opted for a reduction of the information based on the establishment of categories of analysis and selection of relevant associated quotations. The categorization process, which was carried out inductively, allowed the definition of the explanatory hypotheses of the work. These hypotheses emerge with the interpretation of the data and must be assessed as reasonable explanations that can be taken into account in future work.

## 4. Profile and Geography of Digital Nomadism in Spain

As mentioned before, DNs have been conceptualized as a new type of tourist–worker. In Spain, in most cases, they work solely in an online environment, while temporarily living in major cities, natural environments or places where they pursue hobbies. As revealed by

the social networks they use, three distinct profile segments of DNs are displayed. First, the digital nomads stricto sensu, i.e., those people who, by engaging in the knowledge economy through the Internet, are highly mobile. In this segment, different groups can be recognized depending on their level of mobility, age or motivation, although they tend to be young and, in all cases, mobility gives meaning to their way of life.

Second, the segment of nomadic families. This group includes, above all, couples, but households with children are becoming increasingly important. The obligation to send children to school ends up being a constraint on the free mobility of these families, so they tend to be less mobile. Therefore, this segment has a greater incidence in non-school periods (shorter stays) or in longer stays (during an entire academic year).

And finally, corporate nomads and freelancers working remotely. This group is undoubtedly that which has increased the most with the pandemic, according to that published on social media. In this case, the development of teleworking and a greater permissiveness on the part of companies to encourage the relocation of work have led to an increase in the number of remote workers who also relocate their homes for extended periods of time.

The sum of these segments within the group of DNs has made Spain a destination of global importance in the reception of DNs. According to the information from the Nomad List portal, at the end of 2021, there were 23 relevant destinations, a number like that of other surrounding countries such as France or Italy.

In general terms, residential preferences of DNs are the two archipelagos (Balearic and Canary Islands), which include eight consolidated destinations and the Mediterranean coast, where seven areas stand out (Barcelona, Valencia, the Costa del Sol in Malaga, Tarifa and the coast of Alicante). The rest are urban destinations located on the Cantabrian and Atlantic coasts (Bilbao, San Sebastian and La Coruña) and in the interior of the country (Madrid, Seville, Cordoba, Granada and Pamplona). In terms of the number of nomads who have reported having visited each destination in the Nomad List portal, the three most populous Spanish cities stand out: Barcelona, with more than 421,000 DNs; Madrid, with more than 203,000; and Valencia, with just over 84,000. Las Palmas de Gran Canaria (slightly more than 62,000 nomads) and Tenerife (Canary Islands); Malaga and Seville (Andalusia); and Palma de Mallorca (Balearic Islands) (between 42,000 and 48,000 nomads each) also have significance in absolute numbers (see Figure 2).

Therefore, in the geography of the destinations, Spain's main economic centers (Madrid, Barcelona, Valencia, Seville, Malaga and Bilbao), coastal cities with an important tourism specialization (Las Palmas de Gran Canaria, Santa Cruz de Tenerife and Palma), other coastal tourist destinations (Ibiza, Corralejo, Javea, Marbella, etc.) and inland heritage cities (Cordoba, Pamplona and Granada) stand out.

Based on the rating of the destinations' characteristics on the Nomad List portal and the comments of nomads in the analyzed networks, the requirements of a place to be attractive for DNs are of three types: (i) those related to working conditions, (ii) those linked to connectivity and (iii) those concerning quality of life.

Among the first, availability and value for money in collaborative workplaces, high-speed internet and good open connectivity in public spaces are essential. Among the second, the connectivity and accessibility provided by the means of transport between origin and destination stand out. Finally, quality of life is the most complex factor, given the high number of psychological elements that make up the perception of this in each person, especially during the de-escalation, in the pandemic period, when vaccination certificates helped to reduce the fear of unsafe stays, as some studies have shown with regard to the catering sector [50].

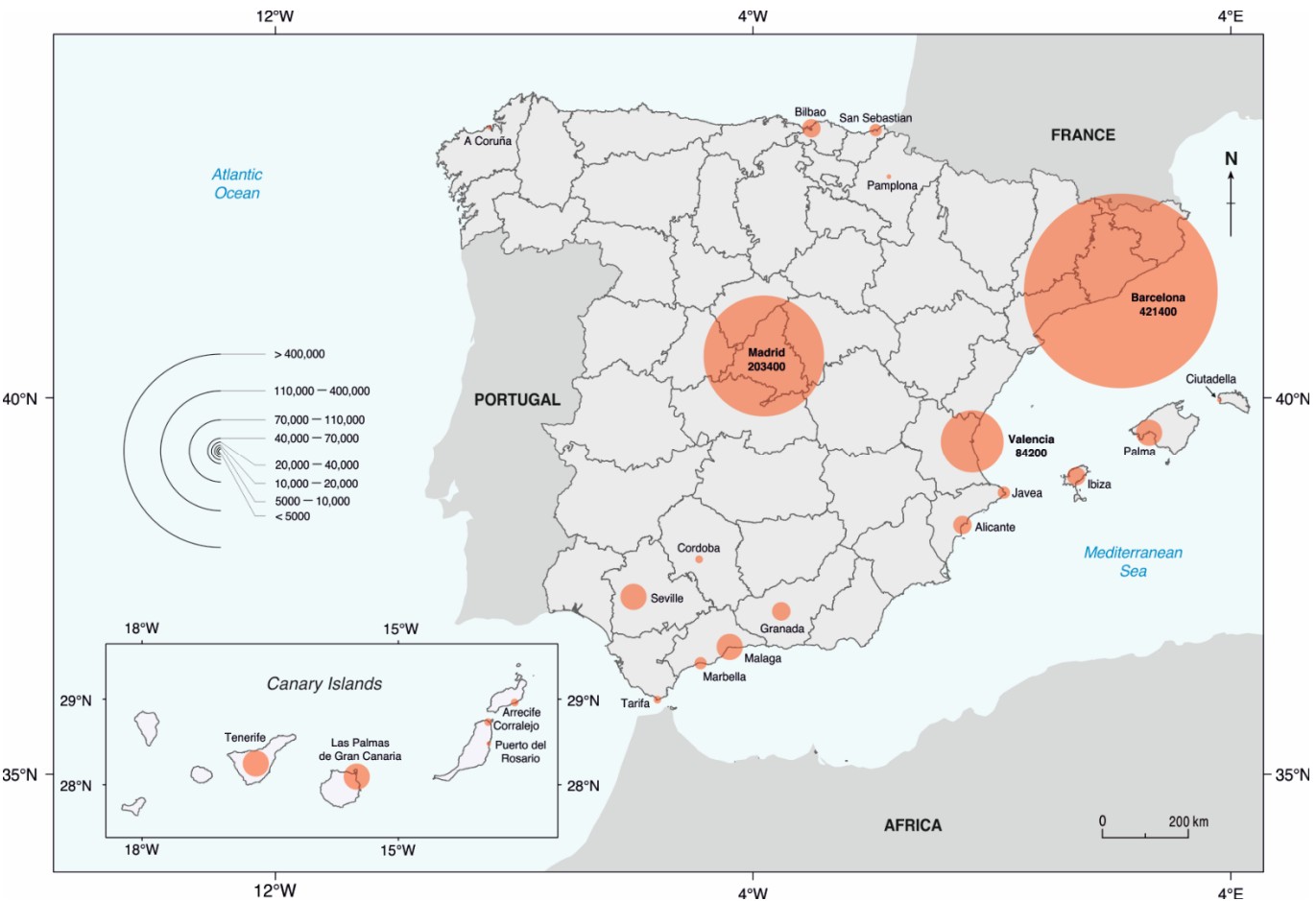

**Figure 2.** Number of digital nomads who have visited Spanish destinations. Source: Nomad List (2021). The authors.

For DNs, the most important criteria explaining the perceived quality are good weather, air quality, safety, a tolerant urban environment for women, foreigners or LGBTQ+, ease of using English in everyday life, low cost of living, feasibility of outdoor leisure activities, cultural resources, possibility of moving around cities on foot and availability and quality of educational and health services. The importance of one or another factor is related to the characteristics, needs and expectations of each person, but the life project they tend to develop in the receiving areas is based on having a home in which to work with a good internet connection and in which they may enjoy their leisure time surrounded by other DNs. Home, virtual communication, leisure and socializing with peers are the key categories explaining the reasons for the temporary change of residence, as can be seen in the following quote:

> "Hi guys! Winter is coming. I want to book some trips to work remotely from GC (. . .) I am looking for a home away from home (. . .) not an apartment in a hotel. Must have fast and stable internet and ideally a pool and BBQ would be nice. Does such accommodation exist? Even better, if it's co-living/co-working in Nomad space." (T.)

Consistently, the perceived quality of life is what seems to influence more in the global evaluation of the destinations on the part of the DNs in the Nomad List Portal, there being a correlation of 0.85 between the perceived quality of life and the ranking that the places have in the 23 Spanish destinations' list.

In addition, the analysis of nomads' statements on social networks leads us to highlight three elements that explain the geographic mobility of DNs: the possibility of outdoor

sports activities, the existence of stereotypical imaginaries and the presence of reasonable costs of living in the destinations.

In relation to the first, some destinations base their success on the existence of a leisure offer linked to the practice of certain outdoor sports activities. For example, Corralejo, in Fuerteventura, and Tarifa, in Cadiz, are two destinations adding the possibility of windsurfing and other water sports all year round to the weather conditions, beaches and a tourist atmosphere [51]. Likewise, the cities of Las Palmas de Gran Canaria and San Sebastian have also become reference destinations for surfers. Las Palmas de Gran Canaria has developed a sports industry in the bay of Confital and Playa de Las Canteras that has made it a World Surfing Reserve (https://www.savethewaves.org/, accessed on 5 December 2022) [7]. For its part, the Zurriola beach, in San Sebastian, is a destination which is famous for its waves.

In the demands for housing to reside temporarily in these four destinations, it is common to make allusion to windsurfing and surfing in social networks and even to specify, sometimes, the desire to share housing with other lovers of both sports.

> "Hello everyone. I and my friend (. . .) are coming to Las Palmas in November. We like to do sports, I surf, we are easy going, we travel a lot, and we can definitely share some experiences from our adventurous life or from the online world. . ." (P.)

Therefore, one of the motives that drives the mobility of DN residents in Spain is the practice of certain outdoor sports in an environment shared with peers. In other words, it is not only about enjoying a hobby, but also about feeling part of a community.

In other cases, the stereotypical imaginary of a place prevails in the choice. This is the case of Ibiza. For this island, two imaginaries can be recognized as elements of attraction: the possibility of carrying out an alternative way of life and of living an experience based on fun. Ibiza began to receive transnational flows at the end of the 20th century thanks to the media impact that implied the development of communities with alternative ways of life and of a hedonic culture that enthroned fun and freedom and that had its most conspicuous manifestation in discotheques and techno music clubs [43]. A good part of the nomads who arrive on the island nowadays maintain this imaginary, claiming the wish of contacting the legacy of the hippie culture and of experiencing the night.

Something similar in relation to imaginaries can be said with respect to cities like Seville, Cordoba or Granada, in Andalusia. These base their attractiveness, not only on their heritage, but in cultural exoticism because of the confluence of Christian, Muslim and Jewish elements, namely, the quintessence of "Spanishness" [52]. Along with this halo of typicality, a bohemianism imaginary has been reinforced with the presence of alternative communities, such as the hipsters in Granada. As a whole, this is a powerful element of recent appeal for DNs who emphasize in their publications the desire to experience authenticity.

Lastly, the geographical mobility of DNs is largely explained by the cost of living. According to Nomad List, the cost of a stay in Spain was USD 2978 per month on average, with large disparities between the most expensive destinations in the northern half of Spain (San Sebastian leading with USD 4753) and the Balearic Islands and the most economical destinations, such as those in the Canary Islands (USD 2126 per month in Tenerife, the lowest figure in the whole country). The best valued destinations tend to be the ones with the lowest cost of living. The same can be said for the length of stay. Its average was 10.7 days in the whole country, but in cities with the lowest cost of living, such as Las Palmas de Gran Canaria, Tenerife, Corralejo or Javea, stays exceeded 25 days. In contrast, in San Sebastian, with the highest cost, the average was only 3 days. Therefore, the cost of living, conditioning the valuation and length of stay of the DNs, is a key element in explaining this geographical mobility.

Besides, given that the cost of living has a positive and high correlation with the price of hotels (0.68) and that of short-term rentals through Airbnb (0.89), it can be concluded that the cost of living is mainly conditioned by the price of accommodation, as can be inferred from NomadList and read in network posts such as this:

"We are a young family with a smiling 8-month-old son, working remotely. I came a few weeks ago to spend the cold season here and decide where exactly to stay. Ready to see any non-agency options (. . .) At least 60 m$^2$ and <900€ (1100€ in case of chalets without agents) Thank you in advance!" (K.)

The frequent reference to the price of accommodation in the networks and the comments in which they highlight the low cost of living in relation to that existing in their countries of origin indicates that we are not dealing with a mobility of the privileged but a mobility in which transnational disparities in the cost of living become a very important factor for attraction. Although it is difficult to deduce that this mobility perpetuates the transnational differences and inequalities [53], what does seem clear is that it is a mobility characteristic of digital capitalism, which benefits from international imbalances in income and wealth.

In conclusion, because of the common geography of destinations of DNs and tourists, it can be concluded that the DNs have a mobile behavior, which is very close to that of tourists [39] and that the cost differential, between the origin and destination areas of both, is a factor of great importance. This parallelism has driven the actions of entrepreneurs and institutions for the attraction of DNs to many Spanish tourist destinations after the first stage of the COVID-19 pandemic, as a strategy to compensate for the decrease in the influx of tourists.

## 5. Policy Agenda and Digital Nomadism

In recent years, destinations around the world have quickly responded to the new phenomenon of digital nomadism and started to market and project themselves as being digital nomad friendly sites [54]. This response has not only entered the political agenda of different governments at the state level, through attractive taxation, visa-free stays, e-residency or digital nomad visas, but also in that of regional and local authorities that have promoted other types of initiative, such as symposiums and conventions of DNs and campaigns to attract and promote their destinations.

From the perspective of the states and therefore in the framework of international telework (i.e., excluding mobile workers, expatriates and cross-border workers), among the initiatives that have been developed, only one was implemented prior to the COVID-19 pandemic. We refer to that adopted by Estonia, the e-Residency. With this measure, the users could establish a company within a day, manage it online from anywhere in the world, open a bank account, have access to online payment providers, digitally sign documents and contracts, and declare taxes online [45]. Following the pandemic, Estonia went further, introducing the digital nomad visa, in June 2020. This allows a maximum stay of one year in the country for "location-independent workers who perform their duties remotely using telecommunications technology" [55].

One month later, in July 2020, the Barbados government introduced a "12-month Barbados Welcome Stamp" to invite remote workers to stay on the island [56]. In the same month, also to overcome the economic hardship due to the coronavirus pandemic, the Georgian government announced a new visa policy that aimed to attract remote workers and the self-employed to live and work there, to stimulate the economy and slowly reopen the borders after the pandemic, in a safe and controlled way [57]. As for 2020, 16 countries opened their borders for DNs, including Mexico, Portugal and Iceland [58]. In January 2021, Croatia joined a digital nomad visa initiative. Finally, Germany has more recently introduced a visa for DNs for up to three years [59].

In 2022, the number of countries with specific visas rose to 30, and in the coming months, the number will continue to increase. In some countries, such as Portugal, the temporary resident visa for freelancers and start-ups was differentiated, opening the possibility of articulating the nomad visa with the permanent residence visa. In all these cases, the worker eligible for one of them is a foreigner who is employed or teleworks for a company or for their own company without the company being registered in the country and without performing any services for employers in the country.

In Spain, to date, most non-EU DNs reside with a tourist visa, which extends up to three months when required (there is a list of countries that do not belong to the Schengen area and whose citizens do not require a visa). For longer stays there is no specific visa, although residence visas for entrepreneurs (REM), for intra-corporate transfer and for highly qualified professionals (HQP), are the tools used by some DNs when they wish to stay for more than three months. In this regard, it should be noted that Spain has specific residence visas for the acquisition of real estate (RIV) or for non-profit residence. These modalities are adapted to other mobility situations, such as wealthy people or retirees, but there is still no specific visa for remote workers and digital nomads.

It is expected that by the end of 2022 the preliminary draft of the Law for the Promotion of the Start-up Ecosystem will have been approved, which contemplates the creation of a new visa for remote workers of foreign companies. In order for a digital nomad to obtain the visa, they must prove that they work remotely for companies based outside Spain; that they are a highly qualified professional and have a minimum professional experience of 3 years [60].

The situation is different for those coming from the European Economic Area (Schengen Area). They do not need a visa for stays of more than three months and can reside as long as they are self-employed or working as an employee, have health insurance and are registered in the Central Register of Foreign Nationals. Currently, most of the DNs that Spain receives come from these countries.

Therefore, we can conclude that the pandemic favored the approval of regulations to encourage the arrival of DNs in many countries. However, in Spain, the stimulus measures for attracting DNs were developed as tourism promotional actions from a regional and local level. One of the best examples was that of the Canary Islands, so we will focus on the analysis of this case study to understand how and why these political actions were launched. The measures developed in the Archipelago can be classified into four different types: (i) information and promotion of destinations; (ii) direct support measures for temporary settlement; (iii) strengthening the professional ecosystem for DNs; and (iv) creation of virtual communities. The measures carried out, the responsible administration and their scope are presented in Table 2.

In relation to the measures for information and promotion of the destinations, the first actions aimed at promoting the Canary Islands as a destination for DNs date back to 2014. In that year, the Gran Canaria Tourist Board and the City Council of Las Palmas de Gran Canaria, in coordination with the Repeople company, created an online platform with this purpose (Nomad City Gran Canaria; https://www.nomadcity.org, accessed on 5 December 2022). With the pandemic, the Island Council of Gran Canaria increased the sponsorship of the publishing of online information and carried out a guide for DNs (2021) [61].

The Island Council of Tenerife and other island institutions together with the Intech company carried out a similar program of actions, under the umbrella of the Tenerife Work and Play brand (https://www.tenerifeworkandplay.com, accessed on 5 December 2022). The initiative focused on the information on accommodation, workspaces, visa procedures, residence permits, the economic identification number for foreigners, the Spanish labor and social security system, public aid for business start-ups, transportation, health, education, learning Spanish and so on.

**Table 2.** Actions to promote the arrival of Digital Nomads to the Canary Islands.

| Measure | Institution | Scope of Action |
|---|---|---|
| Information and promotion of destinations | | |
| Nomad City Gran Canaria | Island Council of Gran Canaria City Council of Las Palmas de G. C. | Island and Local |
| Tenerife Work and Play brand | Island Council of Tenerife | Island |
| La Laguna Digital Nomads | City Council of San Cristóbal de La Laguna | Local |
| "The office with the best climate in the world" Campaign | Canary Islands Government | Regional |
| Fuerteventura Digital Nomads Project | Island Council of Fuerteventura | Island |
| FloWorking La Oliva | City Council of La Oliva | Local |
| Collaboration with long-term rental companies (Live and Work anywhere, Airbnb) | Canary Islands Government | Regional |
| Direct support measures for temporary settlement | | |
| Landing Packages and Welcome Pass | Island Councils | Island |
| Empowerment of professional ecosystem | | |
| Repeople Conference | Island Council of Gran Canaria City Council of Las Palmas de G. C. | Island and Local |
| IX "Lanzarote Creates" Entrepreneurship Conference: Digital Nomads and Remote Workers | Island Council of Lanzarote University of Las Palmas de G. C. | Island |
| Support for co-living companies | Canary Islands Government Island Councils | Regional and Island |
| Tenerife Digital Nomad Ted Nomad | Canary Islands Government Island Councils | Island |
| Nómadas La Palma | Island Council of La Palma Rural Development Association of La Palma (ADER) | Island |
| Creation of virtual communities | | |
| Tenerife Remote Workers and Digital Nomads Meetup, Slack and Facebook | Island Council of Tenerife | Island |
| Floworking La Oliva | City Council of La Oliva | Local |

Source: The authors.

The pandemic has been the reason why the regional government also initiated recruitment promotions for DNs. In 2020, the Canary Islands Government launched a campaign aimed at attracting 30,000 remote workers with the purpose of having them settle on the islands for 1 to 3 months. This action, which had the slogan "The office with the best climate in the world", involved an investment of half a million Euros through the usual means of tourism promotion, media and specialized online spaces such as WIFI Tribe. The Government Department of Tourism reported that the arrival of nomads, despite the pandemic, increased by 10% in 2020, in a situation of zero tourism. In 2021 the increase in DNs was that of 67%.

With a similar purpose, in 2021 the Fuerteventura Tourist Board launched a campaign in the media and social networks with the Fuerteventura Digital Nomads project and with the slogan "Paradise Office". A few months earlier the municipality of La Oliva (including the tourist destination of Corralejo) developed another called FloWorking La Oliva, essentially aimed at floworkers, that is, DNs demanding a quiet lifestyle, contact with the sea and with water sports. In this case, in addition to promotion through the media and online spaces, virtual communities were created. The online communities of both campaigns had just over 600 and 400 members, respectively, at the beginning of 2022.

More recently, actions of promotion have focused on collaboration with accommodation rental companies. For example, the Government of the Canary Islands participates in the Live and Work Anywhere campaign through which Airbnb aims to position itself in

long-stay rentals for DNs. Public action is focused on promoting the Canary Islands and improving the positioning of the island's accommodation offer on this marketing platform.

The second type of action carried out by the administration from the onset of the pandemic has focused on favoring the temporary settlement of DNs by using direct support measures. The most representative case is that of the Landing Packages and Welcome Pass, programs to facilitate arrival that include information and advice for settlement, discounts on various leisure services and the organization of welcome and socialization events.

Thirdly, the institutions have also sought to promote the creation of an ecosystem of professional and business networks. For this reason, they provide direct support to private initiatives that organize events for exchange between agents. In this sense, it is worth highlighting the Repeople Conference. Celebrated since 2016, it is the most relevant annual thematic conference about nomadism in Europe. Besides being a tool of promotion of The Canary Islands, the conference has fostered the development of an island business ecosystem around DNs. In general terms, this and other similar conferences aim to reproduce and reaffirm the nomadic identity, to generate business expectations and to recreate an atmosphere of happiness, optimism and self-realization.

Finally, the creation and dynamization of virtual communities in social networks has also been an objective of public institutions in recent years. Administrations very soon realized the importance that social networks have for a destination, since they become spaces for the exchange of information, services, work and for the socialization of the DNs. So, given the lack of private initiative, some local administrations have launched them. Tenerife Remote Workers and Digital Nomads Meetup, Slack and Facebook, the latter with more than 17,000 members at the beginning of 2022, are successful examples of networks driven by public institutions.

In summary, the political action developed in the Archipelago has tried to attract the DNs through the tourism promotion of the islands and the implementation of concrete measures that make life easier for the DNs and allow them to have an experience according to their expectations. In this sense, aspects such as the possibility of socializing with peers and enjoying leisure in the open air are essential elements.

Therefore, in Spain, similar to a certain extent, in Italy and Portugal [62], the mobility of DNs was treated as an essentially tourist phenomenon during the pandemic because it was intended to temporarily alleviate the decline in tourist arrivals that occurred during the health crisis. However, this circumstantial fact has meant that the mobility of DNs has been consolidated. Part of the tourist accommodation, which had quickly adapted its business to the long-term tourism products, basing its strategy on a reduction of hiring prices and incorporation of services for DNs, has maintained this marketing model after the pandemic. At the same time, since 2021, the amount of new accommodation oriented primarily to DNs—co-livings and apartment buildings—has grown [63].

## 6. Discussion and Conclusions

Digital nomadism, a process of mobility characteristic of an advanced stage of capitalism, has become an issue of increasing interest in academic research. This type of human mobility has been changing at the rhythm of factors like the globalization of services, labor and capital; the development of communications and digital media; and the flexibilization of modes of production and work and lifestyle concepts. Despite their dynamism, these factors were affected by the impact of the pandemic in 2020. However, digital nomadism has gained renewed intensity since the end of confinement, demonstrating its strength.

The studies that have been carried out to characterize this phenomenon have focused on agency, with the sociological and anthropological perspective predominating over the geographical one. This research, however, argues for shifting the academic interest to the places of reception of these remote workers, based on the case studies of Spain and the Canary Islands. In this sense, digital nomadism in Spain has generated a geography of places that overlaps that of tourism (main urban centers, urban-tourist areas, coastal spaces and some inland heritage cities), this being the first research hypothesis. This overlap is a

consequence of the fact that DNs perceive their lives as if they were tourists, selecting their location based on leisure considerations, rather than employment [64].

This is reflected in the scarce interaction they have with the population of the host locations, which only concerns them to the extent that it favors their life project (friendliness, use of English or tolerance in their assessments), which contrasts with their interest in contacting other nomads in the receiving areas, as evidenced by their messages on social networks. Similarly, their comments on destinations are generally focused on the advantages they bring (cost of living, attractions, good weather. . .), reproducing a stereotyped image. The tone of most of the posts is usually positive, and when it is not, they show their displeasure with the services received, just as a tourist would do.

For all of these reasons, DNs reproduce a selective geography of locations, that is, of non-places [65]. Considering Kaplan's statement [66] by which he suggested that the distance that tourists establish with local populations is an element that gives "color" to their experiences, allowing them to maintain their "otherness" as an element of identity, we could consider that DNs also seem to need to differentiate themselves from the host communities to fully enjoy their nomadic experience. It could even be argued that, without this social distance, host locations would not be attractive to them [67]. Moreover, their lack of identity with the host sites leads them to be unaware of the negative repercussions that their mobility can generate. Thus, they mention the increase in the cost of accommodation or the cost of living as elements limiting their life projects, but they do not reflect on the repercussions of their activity on the rise in prices.

Despite the commonalities between tourists and DNs, the latter tend to highlight their differences on the networks. They talk about their travel patterns being more authentic than those of traditional tourism, that their trips do not involve stages of disconnection with their usual lives, that they live experiences in a slow way and that they carry out a life model based on the trip itself as a differentiating factor [68].

The second explanatory hypothesis—that of the mobility of DNs being largely based on income imbalances between countries—can be seen in the geography of nomads' locations in Spain, as well as in their testimonies. By basing their income on the gig economy, the DNs tend to undercut their rates as a competitive strategy to achieve a better position on the Internet. This makes them work below market prices to have positive ratings in their professional profile [69]. Similarly, they are affected by a process of precarization when they work as employees. A part of the fixed business costs, such as that generated by maintaining a job at the company's head office, is transferred to the DNs themselves, without this having an impact on an increase in their income. Therefore, the supposed labor flexibilization of digital capitalism and from which the DNs seem to benefit, turns a good part of them into a social class with downward social mobility, similar to that suffered by other young people in the labor markets of Western countries [70–72].

This is the reason why DNs often choose destinations where their currency or the wage level of their areas of origin gives them comparative advantages. Therefore, the nomads' life project can be interpreted as an adaptive strategy to escape precariousness [73]. The optimistic atmosphere perceived in blogs, conferences and official pages about the business opportunities of the nomadic way of life does not correspond to the difficulties that some reveal. Moreover, we might wonder to what extent digital nomadism causes greater precariousness than the workers would have with a model of life more rooted in their places of origin, due to the underutilization of their family and social capital. Despite this, their mobile behavior allows them to live or feel privileged in relation to the population of the host places. Mobility thus becomes an instrument to live an illusion of privilege, which must be interpreted as a loss of self-criticism and as a factor of distancing and segregation with respect to local communities.

Finally, as the third hypothesis we have considered that the pandemic has consolidated digital nomadism in Spain. Given that this mobility has proved to be more resilient than tourism in the pandemic and that the destinations of nomads and tourists coincide, digital nomadism became a temporary and partial substitute in some destinations during the de-

escalation stage. Some hotels have reoriented their tourism specialization to offer services to DNs while new accommodation spaces, such as buildings of apartments and co-livings, have been created ex profeso. The emergence of this new offer is the best indicator that the pandemic has consolidated digital nomadism.

In many countries, visa policies for DNs (Estonia, Barbados, Georgia, Croatia or Germany, among others) have been the result of the restrictions that the pandemic has caused in other types of mobility. In Spain, the promotion of digital nomadism by local and regional institutions has been much more agile than that of the State, marking a clear difference with what happened on a national level. In the case of the Canary Islands, the actions have been carried out by the regional government and, mainly, by the administrations of each of the islands (Island Councils) as well as some local councils. All of them have been aimed at stimulating the arrival of nomads with information and promotion actions, direct support for settlement, promotion of virtual communities and creation of professional ecosystems.

Beyond the success of these political actions, what has interested us in this article is, firstly, the way in which the measures adopted are in line with the uncritical discourse that surrounds the social and economic fabric of digital nomadism and, secondly, the way in which tourism promotion has adapted to the segments that the digital economy is generating. It can be said that public action has reproduced the optimistic and uncritical discourses that appear in most of the blogs and conferences organized for DNs and in the marketing of companies and professionals that provide their services to this group. In both cases, the aim has been to reinforce feelings of community, messages of personal self-realization and pyramidal business chains as a development strategy. Public action, therefore, has tried to attract the digital nomad through tourism marketing based on these principles, generating expectations that are not always justified. Paradoxically, public action has not bothered to understand the different types of nomads and the territorial implications of the arrival of these social groups. If we take into account that digital nomadism can be considered as a factor that increases income disparities, we can conclude that the attraction policies that have been put in place are favoring these imbalances.

In summary, in the current context of digital capitalism, in times of pandemic, the analysis of the workation migrations in Spain, from the perspective of place and the policies developed, allows us to reflect on the blurred boundaries between tourist, labor and lifestyle mobility, on the relationship between regional socioeconomic imbalances and migrations and on the role assumed by public institutions in the reproduction of these disparities.

**Author Contributions:** Conceptualisation, J.P.-C. and J.D.-M.; Research, J.P.-C., J.D.-M. and C.M.-M.; Methodology, J.P.-C. and J.D.-M.; Writing—original draft, J.P.-C., J.D.-M. and C.M.-M.; Writing—review & editing, J.P.-C. and J.D.-M. All authors have read and agreed to the published version of the manuscript.

**Funding:** This research was funded by MCIN/AEI/10.13039/501100011033 and ERDF-A way of making Europe, grant number RTI2018-093296-B-C21 (Housing and international mobility in cities of the Canary Islands: the emergence of new forms of urban inequality) and by the CANARY ISLANDS GOVERNMENT (Smart Specialization Strategy of the Canary Islands RIS-3) and ERDF-Operational Program, grant number ProID2021010005 (The post-COVID-19 territorial balance in the Canary Islands. New strategies for new times).

**Institutional Review Board Statement:** Not applicable.

**Informed Consent Statement:** Not applicable.

**Data Availability Statement:** Not applicable.

**Conflicts of Interest:** The authors declare no conflict of interest.

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
