# Peer review of "Reflections on Digital Nomadism in Spain during the COVID-19 Pandemic—Effect of Policy and Place"

_sustainability, doi:10.3390/su142316253_

Round 1
Reviewer 1 Report
The title is somewhat misleading, as it should state the fact that there is no empirical/statistical study. So, it is more of a literature/policy review.
I highly recommend that in the introduction you should only state the hypothesis, and avoid arguments for or against them, as these arguments should pe presented in the following chapters.
Reviewer 2 Report
The authors investigated the current issue of the increased influence of digital technologies during the pandemic. The authors call digital influences nomads, which is very interesting in showing their strength during the health crisis. They follow the structure of writing the manuscript according to the instructions, the abstract contains all the necessary structural elements. In the introductory part, they present the term digital nomads through the period of global capitalism, but also as a lifestyle of the new age. They tabulated the results for social networks of digital nomads consulted. In the introductory part and the review of the literature, the aim, essence and importance of the research can be clearly seen. Regardless of the fact that statistical models are not used, research of this type is very interesting. The authors have provided a tabular representation for actions to promote the arrival of Digital Nomads to the Canary Islands. The discussion and closed considerations show
on the one hand, that it is not a flow of privileged people but a mobility like tourism related to the difference in international incomes. On the other hand, these results
points out that the consolidation of digital nomadism during the pandemic is linked to the tour. It is suggested to revise the English translation, check the structure of writing the entire manuscript according to the journal's instructions, from the introduction to the references, and use the recommended literature:
Gajić, T., Petrović, M., Blešić, I., Vukolić, D., Milovanović, I., Radovanović, M., Vuković, D., Kostić, M., Vuksanović, N., & Malinoivć, S. (2022). COVID-19 Certificate as a Cutting-Edge Issue in Changing the Perception of Restaurants’ Visitors – Illustrations From Serbian Urban Centers. Front. Psychol. 13:914484, 1-11.doi: 10.3389/fpsyg.2022.914484
If the authors do not wish to conduct statistical or experimental research, it is recommended that the manuscript be accepted as a single study or review article.
Reviewer 3 Report
The manuscript entitled "Digital nomadism in Spain: policy and place during the COVID-19 pandemic" is very interesting, and the issues it deals with are current. However, it still has some issues that need to be addressed. Below is the list of suggestions for manuscript enhancement:
- The paper's aims are not clearly defined. The authors suggested three hypotheses for which they provided insufficient justification. Each of the presented hypotheses must be backed by data and sources pertinent to the study question. The authors' formulation of hypotheses is vague. For example, the authors claim that " The last hypothesis suggests that the pandemic contributed to the consolidation of digital nomad flows in Spain, replacing, in part and temporarily, international tourists..." What is the basis for this hypothesis? It is very important to list the sources that the authors used to make their assumptions.
- The research methodology is insufficiently described. The authors noted that their research is based on qualitative research that was not presented in the article. Therefore, the authors listed three hypotheses from which it was impossible to infer scientifically verified facts and conclusions. Qualitative research entails the collection and analysis of no numerical data (e.g., text, video, or audio) to comprehend thoughts, opinions, or experiences. It can be used to get an in-depth understanding of a subject or to develop fresh research ideas. Each of the study methodologies employs one or more data collection techniques. These are some of the most prevalent qualitative techniques: Observations: documenting in detail what researchers have observed, heard, or encountered; Individually posing questions to individuals in one-on-one talks; Focus groups: asking questions and encouraging conversation among a group; Surveys: distributing questionnaires with open-ended inquiries; Secondary research is the gathering of existing information in the form of texts, photos, audio or video recordings, etc. For this reason, it is essential, when writing up the methodology for qualitative research, to reflect on a strategy and explain in detail the decisions made during data collection and analysis. For this research, in particular, a good method would be to collect information already out there in the form of texts, which should then be analysed using some of the free software for textual data analysis.
- The submitted research technique and results are based on descriptive indicators, which are insufficient for the prestigious journal Sustainability. I recommend that the authors specify the objectives of the research more precisely and utilise one of the qualitative research methodologies to demonstrate the results that will have scientific importance and be applicable to various fields.
- Please indicate the source of the data (Lines 91 and 92)
Round 2
Reviewer 3 Report
The authors revised the manuscript based on the reviewers' instructions and comments. In my opinion, the paper is suitable for publication in the scientific journal Sustainability.